# Why Venous Leg Ulcers Have Difficulty Healing: Overview on Pathophysiology, Clinical Consequences, and Treatment

**DOI:** 10.3390/jcm10010029

**Published:** 2020-12-24

**Authors:** Joseph D. Raffetto, Daniela Ligi, Rosanna Maniscalco, Raouf A. Khalil, Ferdinando Mannello

**Affiliations:** 1Vascular Surgery Research Laboratories, Division of Vascular and Endovascular Surgery, Brigham and Women’s Hospital, Harvard Medical School, Boston, MA 02115, USA; raouf_khalil@hms.harvard.edu; 2Department of Biomolecular Sciences, Section of Biochemistry and Biotechnology, Unit of Clinical Biochemistry, University Carlo Bo of Urbino, 61029 Urbino, Italy; daniela.ligi@uniurb.it (D.L.); r.maniscalco@campus.uniurb.it (R.M.)

**Keywords:** chronic venous disease, chronic venous insufficiency, venous leg ulcer, healing, biochemistry, pathophysiology, clinical medicine, therapy

## Abstract

Venous leg ulcers (VLUs) are one of the most common ulcers of the lower extremity. VLU affects many individuals worldwide, could pose a significant socioeconomic burden to the healthcare system, and has major psychological and physical impacts on the affected individual. VLU often occurs in association with post-thrombotic syndrome, advanced chronic venous disease, varicose veins, and venous hypertension. Several demographic, genetic, and environmental factors could trigger chronic venous disease with venous dilation, incompetent valves, venous reflux, and venous hypertension. Endothelial cell injury and changes in the glycocalyx, venous shear-stress, and adhesion molecules could be initiating events in VLU. Increased endothelial cell permeability and leukocyte infiltration, and increases in inflammatory cytokines, matrix metalloproteinases (MMPs), reactive oxygen and nitrogen species, iron deposition, and tissue metabolites also contribute to the pathogenesis of VLU. Treatment of VLU includes compression therapy and endovenous ablation to occlude the axial reflux. Other interventional approaches such as subfascial endoscopic perforator surgery and iliac venous stent have shown mixed results. With good wound care and compression therapy, VLU usually heals within 6 months. VLU healing involves orchestrated processes including hemostasis, inflammation, proliferation, and remodeling and the contribution of different cells including leukocytes, platelets, fibroblasts, vascular smooth muscle cells, endothelial cells, and keratinocytes as well as the release of various biomolecules including transforming growth factor-β, cytokines, chemokines, MMPs, tissue inhibitors of MMPs (TIMPs), elastase, urokinase plasminogen activator, fibrin, collagen, and albumin. Alterations in any of these physiological wound closure processes could delay VLU healing. Also, these histological and soluble biomarkers can be used for VLU diagnosis and assessment of its progression, responsiveness to healing, and prognosis. If not treated adequately, VLU could progress to non-healed or granulating VLU, causing physical immobility, reduced quality of life, cellulitis, severe infections, osteomyelitis, and neoplastic transformation. Recalcitrant VLU shows prolonged healing time with advanced age, obesity, nutritional deficiencies, colder temperature, preexisting venous disease, deep venous thrombosis, and larger wound area. VLU also has a high, 50–70% recurrence rate, likely due to noncompliance with compression therapy, failure of surgical procedures, incorrect ulcer diagnosis, progression of venous disease, and poorly understood pathophysiology. Understanding the molecular pathways underlying VLU has led to new lines of therapy with significant promise including biologics such as bilayer living skin construct, fibroblast derivatives, and extracellular matrices and non-biologic products such as poly-N-acetyl glucosamine, human placental membranes amnion/chorion allografts, ACT1 peptide inhibitor of connexin 43, sulodexide, growth factors, silver dressings, MMP inhibitors, and modulators of reactive oxygen and nitrogen species, the immune response and tissue metabolites. Preventive measures including compression therapy and venotonics could also reduce the risk of progression to chronic venous insufficiency and VLU in susceptible individuals.

## 1. Introduction and Scope of the Problem

Venous leg ulcer (VLU) is the most common type of ulcer in the lower extremity [1]. VLU accounts for 70–80% of ulcers presenting for evaluation and treatment to different professions across different specialties including primary care physicians, geriatricians, wound care specialist, phlebologist, surgical specialties, cardiologist, and vascular surgeons. The prevalence of VLU is up to 2% of the population and, importantly, increases to 5% of individuals over the age of 65 years old [2,3]. Venous leg ulcer is a worldwide problem in many countries and regions including the United States, the United Kingdom, Australia, India, Africa, and Europe. The number of affected individuals is staggering in Africa, with an estimated 25 to 135 million individuals having VLU and chronic wounds (with the majority of them being VLU). Europe has up to 2.2 million people affected, and over 6 million individuals are affected in the United States [4]. It is important to note that VLU can heal with good wound care and compression, which is the mainstay of treatment. Healing rates of VLU of 76% at 16 weeks can be achieved with compression [5]. However, a major issue with VLU is the high recurrence rates, which can be significant and as high as 50–70% at 6 months [1]. The morbidity of VLU has many financial and socioeconomic impacts, especially given the high recurrence rates. The treatment of VLU is significant, involving and requiring many resources, specialties, appointments, inconveniences to the patients, wound care products, psychosocial events, and hardships and has a major healthcare burden. After taking into consideration all aspects of caring for patients with VLU, including doctors visit, nursing care, wound care, and bandages applied along with compression; surgical and endovenous treatments; and hospitalization for complications related to pain, drainage and progression, and infections, the cost becomes exponentially elevated. The associated costs for VLU care are just over $15,000 but increase significantly for patients who have delayed healing and can result in costs as high as $34,000 per patient per year, with most of the cost driven by outpatient visits, nursing care, and admissions to hospitals for related complications, usually infection [1,6].

Patients with VLU have increased missed workdays, with 29% higher work-loss costs. However, a price on the burden endured by patients with VLU cannot be estimated when one takes into account the psychosocial impact with significant isolation, embarrassment, negative emotions, anxiety, depression, loss of self-worth, dependency, and sleep disturbance. The annual United States taxpayer burden for VLU is estimated at an astonishing cost of $14.9 billion (Figure 1) [7]. 

The intention of this comprehensive review is to provide practitioners caring for patients with VLU with a foundation of information that will define causes of VLU and other ulcers that are less common that may be mistaken for VLU, the clinical manifestations of VLU, and delayed healing of and difficulty healing VLU that is common place in clinical practice and to provide pathophysiological molecular insights on important regulators and inflammatory mediators that are critical factors in propagating the VLU refractory state of continued inflammation, surgical treatments and innovations, and drug therapies that have evolved given our increased scientific discovery and knowledge that lead to better targeted therapies and finally with information on the means to prevent progression, occurrence, and recurrence of VLU.

## 2. Pathophysiology

### 2.1. Definition and Etiology of Venous Leg Ulcers

VLU can be defined as a full-thickness defect of the skin frequently seen in the ankle region that fails to heal spontaneously and is sustained by chronic venous disease (CVD, the spectrum of venous diseases affecting the lower limbs) [8]. In more recent guidelines, a VLU is defined by best practice and uses the standard definition of an open skin lesion of the leg or foot that occurs in an area affected by venous hypertension [1]. 

VLU is a complex system involving mechanisms that affect venous macrovasculature and microvasculature. The macrovasculature involves abnormalities with hemodynamics, leading to venous hypertension that involves superficial venous insufficiency that can overwhelm the deep system, junctions, and reentry points in compartments of the lower extremity and cause outflow obstruction via the iliofemoral venous system, calf muscle pump dysfunction, and perforator venous insufficiency. The majority (70–80%) of patients with VLU have primary venous insufficiency (reflux) from varicose vein disease, and about 20–30% have secondary venous insufficiency from post thrombotic syndrome (PTS) [9]. Although there are many more patients with primary venous insufficiency, PTS has a much higher risk of developing VLU and is much more aggressive in its natural history, making treatment more challenging [10]. The microvasculature, which includes the glycocalyx and endothelium, is affected by changes in shear stress and activation of leukocytes and adhesion molecules occurring in both large and microscopic veins. The microvascular system is composed of a network of capillaries, post-capillary venules, interstitium, and lymphatics that respond to overexpressed inflammatory pathways and upregulation of cytokines, chemokines, matrix metalloproteinases (MMPs), iron free radicals, and activated oxygen and nitrogen species that all have detrimental effects to the surrounding tissues and possibly systemic effects (Figure 2). 

Both the macrovenous and microvenous components of the venous system are affected. In the macrovenous component, there are several abnormalities including venous valve dysfunction and obstruction, that have a common pathway leading to venous hypertension and skin changes including venous leg ulcers. Leukocytes and matrix metalloproteinases (MMPs) have a direct involvement with the pathology seen in venous structures (indicated by bidirectional arrows). In microvenous circulation endothelial dysfunction, glycocalyx injury, and activation of chemokines (e.g., MCP-1 and MIP-1), adhesion molecules (e.g., ICAM-1, VCAM-1, and selectins) and endothelial regulators (NO) are potent molecules to allow for leukocytes migration within the venous wall and valve and eventually in the interstitium. In addition, through oxidative stress (oxygen and nitrogen reactive species), iron activation and innate immunity receptors and their ligands lead to further expression and activation of leukocytes activity (macrophage (MP), mast cells (MC), and T-lymphocytes (TL)). A variety of cytokines are expressed by leukocytes, with both direct and indirect effects, leading to a continuous proinflammatory and inflammatory environment in addition to the proteolytic activation of matrix metalloproteinases (MMPs), which have both been demonstrated to cause endothelial-smooth muscle relaxation, venous wall dilation, proteolytic degradation, and wound formation in venous leg ulcers. Cellular (endothelial cells, smooth muscle cell, and fibroblasts) metabolic changes occur, leading to loss of integrity of the venous wall and valves, that is directly linked with microcirculation resulting in venous hypertension (indicated by the bidirectional arrows). 

A clear understanding of inflammatory pathways allows for detailed understanding of the pathophysiology and for areas of research for treatment targets. In addition, there are significant metabolic changes that occur in the VLU cell and tissues, which affect cell function and potential for healing and also present systemically, indicating that metabolic changes are dynamic and opportunity for novel therapeutic targets [11,12,13,14]. 

### 2.2. Leg Ulcer Differential Diagnosis and Misdiagnosis 

The development of leg ulcers is a clinical sign shared by many diseases. Leg ulcers usually occur in the lower leg or in the foot, with a predominance of venous ulcers located in the gaiter region, near the skin area affected by lipodermatosclerosis or white atrophy [15], and non-venous ulcers in the foot area.

Chronic wounds of the lower extremities could be sustained by several local and systemic causative factors, leading to a broad comparison among ulcers.

It has been estimated that the venous origin impacts 50–75% of chronic leg ulcers, and this percentage heavily increases if foot ulcers are excluded. These numbers are strictly linked to the fact that signs of CVD (i.e., varicose veins, edema, and skin changes) could be observed in at least 25% of the population, thus increasing the probability to diagnose CVD (chronic venous disease )/CVI (chronic venous insufficiency) also in patients affected by other forms of ulcer [15].

Besides the venous origin, other common etiologies are arterial (5–10%), mixed (arterio-venous), neuropathic, diabetic, and pressure ulcers, for which the prevalence reflects overall population aging. 

Table 1 shows the major characteristics of leg ulcer of vascular etiology. 

The location of the wound may help with differential diagnosis. In fact, VLUs are usually located in the gaiter region and exhibit signs of venous CVI (e.g., edema, dermatitis, lipodermatosclerosis, hyperpigmentation, or white atrophy); arterial ulcers are mainly located in the distal regions of the extremities. Pain, sensation of coldness, and changes in skin color following leg elevation usually accompany arterial ulcers [16]. Diabetic ulcers are frequently observed in more distal areas of the extremities (e.g., the lateral or pretibial aspects of the leg, the dorsum of the feet, the malleoli, and the distal aspects of the forefeet and toes); neuropathic ulcers in diabetic patients occur in the plantar area [16].

A broad spectrum of wounds mimics common VLU, and unusual wounds are often misdiagnosed due to concurring risk factors. Accounting for 10% of the leg ulcers, other causes include infections, skin cancers, metabolic disorders, inflammatory processes, and other diagnoses (Table 2) [17].

Several disorders of metabolic, hematological, autoimmune, and connective tissue origin should be taken into consideration during diagnostic workup. Furthermore, neoplastic diseases (e.g., basal cell carcinoma and squamous cell carcinoma) could also result in leg ulceration, and chronic wounds may undergo malignant transformation (Marjolin’s ulcer) [16]. Atypical wounds require histological assessment to be properly diagnosed [17]. The recommendations are that a VLU that has been present for 4–6 weeks or longer should undergo biopsies of the skin edges to evaluate for other possible pathologies, especially if the leg ulcer does not improve with wound and compression therapy and atypical ulcers appear [1].

A careful differential diagnosis is imperative to make the best therapeutic choice because specific treatments depend on the underlying cause. The major challenge is to find the main cause of non-venous leg ulcers in patients where CVD symptoms exist, given the high prevalence of venous disorders in the population. On the other hand, leg ulcers without signs of CVD should be addressed as non-venous ulcers [15].

Misdiagnosis of a leg ulcer has a great impact both on patient’s suffering, due to delayed wound healing, and on economic costs. Furthermore, improper treatments lead to relevant risks including aggravation of the underlying disease, masking of symptoms, delaying appropriate diagnosis, and increasing morbidity or mortality.

### 2.3. Clinical Manifestations, Healing, and Consequences

CVD includes a spectrum of clinical manifestations ranging from telangiectases and reticular veins to skin changes, such as lipodermatosclerosis and VLU. Varicose veins are among the first clinical evidences of CVD. They are enlarged superficial veins that progressively become twisted and dilated. Edema is the first sign of CVI. It appears as fluid accumulation starting from the perimalleolar ankle area to the upper side of the leg. Skin changes, due to red blood cell extravasation, hemosiderin deposition, iron overload, and inflammatory and fibrotic processes, are represented by hyperpigmentation, eczema, atrophie blanche, and lipodermatosclerosis [2].

VLU is the result of the pathological changes developed inside vessels after a prolonged condition of CVI. As the culminating complication of CVI, VLU is accompanied by several clinical manifestations of the underlying disease.

VLUs commonly present as open lesions generally confined in the lower aspect of the leg at the gaiter region extending from midcalf to approximately 1 inch below the malleolus. Wounds can be single or multiple, mainly with an irregular shape and shallow. However, VLU have also been diagnosed and described in unexpected regions including the pretibial, dorsum of the foot, and rarely the toes. The wound skin is characterized by a red granulation tissue with yellow fibrinous tissue on the basis of its healing status; black necrotic tissue rarely occurs. Variable odor and release of exudate can be observed depending on the degree of leg edema or the presence of bacterial colonization, both contributing to delayed wound closure through a decreased supply of nutrients and oxygen to the tissues and a chronically sustained inflammatory response [2]. 

Other clinical features of CVI are generally present, including varicose veins, edema, dermatitis, telangiectasias and reticular veins, hemosiderin pigmentation, lipodermatosclerosis, and atrophie blanche. These clinical manifestations provoke patient’s suffering, swelling, leg pain, pruritus, pain, or nocturnal cramps [18,19].

Normal healing of acute wounds usually proceeds through orderly and time-limited reparative processes (i.e., hemostasis, inflammation, granulation, and remodeling phases) that promote the restoration of the anatomic and functional integrity of the skin. On the contrary, chronic wounds (e.g., VLU) are usually arrested in a prolonged inflammatory phase, thus blocking progression toward the next phases and preventing wound closure [20].

In this respect, according to CEAP (Clinical, Etiological, Anatomical, Pathophysiological) classification, VLU can be classified as healed (C5) or non-healed (C6) ulcers depending on how long an ulcer persists without any improvement. In particular, non-healing VLU is used to define a wound that did not reduce in size within 6 months; both ulcers are blocked in the inflammatory phase of wound healing (inflammatory ulcers), and ulcers which enter the granulation phase but did not reduce in size (granulating ulcers) could be defined as non-healing VLU.

Chronic VLUs provide a fertile breeding ground for the onset of several complications, ranging from immobility and reduced quality of life to cellulitis, severe infections, osteomyelitis, and neoplastic transformation [21].

In these cases, ulcers which persist for long periods of time require biopsy assessment for malignant evolution; moreover, radiography, bone scanning, and bone biopsy are needed if osteomyelitis is suspected.

On the other hand, CVI itself is a great source of complication, including thrombophlebitis, deep vein thrombosis (DVT), pulmonary embolism, and PTS [19]. 

### 2.4. Pathophysiological Mechanisms

VLU is the result of an intricate series of pathological events involving hemodynamic, cellular, and biomolecular alterations of macro- and microcirculation, which are eventually transmitted to the skin. In this complex picture, the common finding is the presence of ambulatory venous hypertension. 

Several predisposing factors (e.g., advanced age, female sex, genetic predisposition, family history, pregnancy, estrogen levels, obesity, prolonged standing, sitting, and environmental/occupational factors) have been highlighted to promote venous hypertension [22]. 

From an etiological point of view, CVD can be the result of congenital, primary, or secondary disorders. Genetic predispositions (e.g., Klippel–Trenaunay, Park–Weber, and Ehlers–Danlos syndromes; CADASIL and FOXC2 gene mutations; and desmulin dysregulation), despite being present at birth, manifest with clinical significance later in life. Other patients without congenital disorders could be affected by primary CVD, and damages to the vein wall and valves could appear before the development of clinically recognized venous hypertension [23,24]. On the other hand, the presence of other acquired conditions (e.g., venous obstruction) could predispose to the development of a secondary venous insufficiency [2].

According to CEAP classification, the pathogenic mechanisms of CVD, the starting point for the occurrence of CVI and VLU, include venous reflux, obstruction, or both. 

Retrograde blood flow or venous reflux in the superficial, perforator, and deep veins is a common feature in patients affected by VLU. The main cause of venous reflux is the presence of venous valve incompetence of several districts (axial deep or superficial veins, perforator veins, and venous tributaries) as well as alterations of hemodynamics and vessel walls, and the imbalance of inflammatory and proteolytic pathways. However, whether valvular incompetence or inflammatory changes within the venous wall and dilation are the cause or the consequence remains a matter of discussion [11].

Valvular incompetence in the superficial venous system and associated reflux have been detected in about 90% of patients with CVD and 84% of patients with VLU [9,25]. Prolonged venous distention, weak vessel walls or leaflets, injuries, or superficial phlebitis are the main causes of valvular incompetence in the superficial systems [2].

Valvular incompetence of deep veins usually results from previous deep vein thrombosis and has been associated with an increased risk of the disease toward ulceration [26,27]. 

Moreover, venous valve incompetence may also occur in perforator veins, thus exacerbating the hemodynamic abnormalities of the superficial system. 

Venous reflux in deep veins is a common finding in patients suffering from primary or secondary CVD, whereas venous obstruction is characteristic of other conditions (e.g., deep vein thrombosis, post-thrombotic syndrome, and venous stenosis) [23]. 

Venous hypertension is further aggravated by failure of the calf muscle pump to move deoxygenated blood from the venous system, which often occurs with severe reflux or obstruction and represents a relevant risk factor for VLU development [28]. 

Venous hypertension in association with the onset of cellular, molecular, and hemodynamic alterations in the microcirculatory system, through the activation of a cascade of events involving inflammatory processes, proteolytic activity, reduction of the physiological shear stress, and loss of glycocalyx glycosaminoglycans, leads to venous structural changes which finally exacerbate venous hypertension resulting in clinical manifestations of CVD, skin changes, and VLU [24,29,30]. 

In fact, the pooling of venous blood due to valve dysfunction together with increased venous pressure on the vein walls lead to an alteration of the physiological shear stress, which normally maintain blood fluidity and inhibit blood cell attachment. These mechanical stress forces alter the endothelium integrity both by disrupting the protective glycocalyx layer and by promoting endothelial cell fenestration/activation. Endothelial cell activation, through the expression of adhesion molecules (e.g., ICAM-1, VCAM-1, and E-selectins) and the release of chemoattractant molecules favor white blood cell (WBC) recruitment, attachment, and migration within the vein wall and interstitial tissue. Once activated, leukocytes and endothelial cells release a plethora of inflammatory and proteolytic mediators, growth factors, and chemotactic signals which synergistically target fibroblasts, vascular smooth muscle cells (VSMCs), and the extracellular matrix (Figure 2) [31].

As a consequence, VSMCs proliferate and lose their contractility and their ability to synthesize collagen fibers, thus resulting in the appearance of hypertrophic areas, with reduced contractility, increased rigidity, and impaired elasticity, which altogether worsen the ability of the vein wall to respond to increased venous pressures and to preserve the physiological shape [32,33].

Similarly, fibroblasts are also affected by altered collagen synthesis and reduced cellular proliferation due to an abnormal response to TGF-β1 signaling and senescence [34]. 

Histological and structural studies have demonstrated that the vessel wall of varicose veins presents regions with decreased collagen content alternated with areas of increased collagen and reduced elastin and laminin [35,36] which contribute to the tortuosity and rigidity of VVs (Varicose Veins). Interestingly, an inverse ratio of collagen Type I to Type III, with an abundance of Type I in varicose vein wall structure, and loss of elasticity due to decreased collagen type III, events regulated by posttranslational modifications likely by MMPs (e.g., MMP-3), have been demonstrated [37,38].

The increased permeability of endothelial cells leads also to the extravasation of red blood cells, the degradation of which within the interstitium entails the release of hemoglobin and ferric iron, which amplify oxidative stress and inflammation of the surrounding tissues, further impairing wound healing [39,40].

In this complex network of hemodynamic, cellular, and molecular processes, proteolytic enzymes and, in particular, the members of the MMP family, released by infiltrating leukocytes as well as by resident fibroblasts, VSMC, and keratinocytes, regulate both pathological remodeling of the extracellular matrix and the availability of signaling molecules. Besides their direct proteolytic activity against extracellular matrix (ECM) proteins, proteoglycans, and glycocalyx glycosaminoglycans, they also modulate inflammatory pathways by processing chemokines, cytokines, and cell surface receptors. In fact, MMPs can activate inactive precursors of pro-inflammatory cytokines; degrade growth factors and receptors; and contribute to magnifying the proinflammatory, degradative, and prothrombotic microenvironment that leads to leukocyte activation and release of other proinflammatory cytokines. 

These mechanisms have been confirmed by a variety of experimental observations. Starting from the original leukocyte trapping hypothesis [41,42], even more evidences have highlighted that blood returning from the feet of CVD patients has reduced white blood cell count [43]. This was also confirmed by histological studies of skin biopsies where increased levels of T lymphocytes and macrophages have been observed [42]. Taken together, these studies indicate the importance of leukocytes in the pathophysiological process of CVD and VLU, and the events of WBC activation and attachment to the endothelium leading to inflammatory processes and disease progression. 

Despite conflicting results that have been occasionally reported, the impact of inflammatory and proteolytic mediators has been widely documented by a number of preclinical and in vitro studies. In this respect, circulating biomarkers have been found both in blood samples and VLU exudate [12,44,45,46,47]. 

Several trigger mechanisms have been argued for CVI and VLU, including fibrin cuff formation, growth factor trapping, and white blood cell trapping. Recent studies proposed that CVD could be considered primarily a blood pressure-driven inflammatory disease, although the chronological sequence of events still remains a matter of debate [22].

However, a comprehensive theory of the pathophysiological mechanism remains speculative and future studies are needed to deepen the knowledge on VLU development.

### 2.5. Biomarkers and Implications for Translational Research and Clinical Practice

The VLU microenvironment is a dynamic milieu where an intricate network of signaling systems exist that include different cells, growth factors, inflammatory and chemotactic mediators, their receptors and downstream signaling molecules, extracellular matrix molecules, proteases, and inhibitors. However, due to the dysregulation of VLUs, VLU are not able to enlist the normal orchestrated wound healing steps that require a series of timely and spatially controlled events involving hemostasis, inflammation, proliferation, and remodeling [48]. During each phase, cells such as leukocytes, platelets, fibroblasts, vascular smooth muscle cells, endothelial cells, and keratinocytes release extracellularly a wide variety of biomolecules (e.g., growth factors, cytokines, chemokines, proteases, proteins) that overall lead to moving wound healing toward the next step. 

Every alteration of the cellular and biochemical components driving the physiological progression to wound closure could represent a factor delaying ulcer healing. Consequently, identifying both mediators of physiological and pathological processes represents a crucial point for research on biomarkers of disease. 

In fact, taking into consideration the definition of a biomarker as “a characteristic that is objectively measured and evaluated as an indicator of normal biological processes, pathogenic processes, or pharmacologic responses to a therapeutic intervention” [49], the research of clinical biomarkers for VLU should emphasize several aspects. It is important to focus attention on diagnostic/screening biomarkers, of which recognition could help to confirm a diagnosis or may be useful in the early diagnosis of patients predisposed to developing advanced stages of CVI, such as VLU; on prognostic biomarkers, which are needed to monitor and predict the progression of the disease; and on predictive/stratification biomarkers for determining treatment benefit and potential for healing, which are also able to identify patients at high risk of developing adverse events (e.g., after pharmacological treatments) and to better guide clinicians to prescribe even more personalized medicine. 

Biomarkers of VLU can be expressed in tissues or fluids or can originate from imaging techniques, or chemical and physiological determinations. To date, the clinical utility of biomarkers has been explored solely in clinical trials and laboratory research. Notwithstanding several studies on physio-pathological mediators reflecting the biological activities occurring within the venous leg wound (reviewed in [2,12,50]), up to now, no biomarkers of clinical biochemistry has been integrated as diagnostic/prognostic/therapeutic tool/panel to the current vascular clinical practice.

Wound healing status is currently evaluated through measurement of the wound area [1]. However, this method is time-consuming and requires several weeks of determinations to discriminate a healing VLU from a non-healing one. Moreover, it delays the choice of a more appropriate and effective management strategy [51,52].

Numerous experimental studies have been performed to monitor disease progression by studying panels of biomarkers ideally discriminating between healing and nonhealing chronic VLU through analysis of the blood, wound fluids, and tissues (reviewed in [53,54,55,56,57,58]). 

Histological studies have demonstrated that chronic venous ulcers are sustained by prolonged inflammatory phase, in which macrophages, neutrophils, and T lymphocytes represent the predominant cell types. This is associated with an increased expression of adhesion molecules, such as ICAM-1, VCAM, LFA-1, and VLA-4 [59,60]. 

A frequent histological finding is represented by fibrin cuff and deposition of actin and collagen IV and by extravasation of factor XIIIa and α2-macroglobulin [59]. 

An increased proteolytic activity has been also observed in non-healing ulcers, mainly sustained by high levels of neutrophil elastase, MMPs, urokinase-type plasminogen activator (uPA), and extracellular MMP inducer (EMMPRIN and CD147) and decreased activity of tissue inhibitors of MMPs (TIMPs). Among growth factors, the TGF-β family has been extensively investigated in wound healing, despite conflicting results reported.

A list of tissue biomarkers found in the ulcer microenvironment is summarized in Table 3.

It is important to note that VLU can heal with good wound care and compression, which is the mainstay and the golden standard of treatment. Faster healing rates of VLU can be achieved with compression that is able to significantly modify several inflammatory biomarkers [87].

On the contrary, wounds that physiologically heal do not show fibrin cuffs, whereas positive immunostaining for all three TGF-b isoforms and type I and type II receptors was observed; furthermore, proteolytic activity can be detected also in healing wounds, which is directed to the remodeling phase [12,53,54,55,56,63,77,106]. The healing of a superficial wound requires many factors to work in concert, orchestrating and balancing a plethora of pro- and anti-inflammatory molecules as well pro- and anti-proteolytic enzymes other than growth factors and signaling chemokines, in an intricate network involving the biochemistry of the ECM and limiting the barriers of both infection and hypoxia (reviewed in [12,22,29,50,51,53,56,57,58,104,106]). 

Overall, histological studies have highlighted that chronic wounds are characterized by a dysregulated healing process with an aberrant distribution of growth factors, cytokines, and enzymes within the wound instead of a reduced cellular activity [59].

However, histological studies, despite being highly informative, are obtained through invasive procedures that reflect a single time point during wound healing, making it difficult for repeated sampling from the same site.

Thus, recent research has focused attention on soluble biomarkers which can be measured with less invasive technique in blood and wound exudate. 

A list of soluble proteolytic biomarkers found in the VLU microenvironment is summarized in Table 4.

Noteworthy, VLU can significantly improve the healing process with good wound care and compression, which is recognized as the golden standard of treatment. In fact, faster healing rates of VLU can be achieved with compression, through a significant modulation of several proteolytic biomarkers [120].

The final goal of experimental studies on VLU biomarkers is to be incorporated into the evaluation of wound duration and area in the clinical practice for wound assessment. 

This is of importance for improving the ability of standard wound measurements to correctly diagnose VLU. In fact, changes in wound surface reflect the cellular, molecular, and biochemical processes occurring in underlying tissue, which start advanced compared to the observation of a decreased wound area. Thus, identifying crucial mediators of wound progression could predict ulcer fate before the appearance of visible changes.

A further challenge is the application of a clinically useful panel of biomarkers in laboratory medicine practice through rapid and inexpensive procedures. In this respect, it is desirable for the development of a predictive test to monitor the healing status of a wound which takes into consideration a combination of wound biomarkers of the healing and non-healing conditions.

## 3. Clinical Aspects

### 3.1. Recalcitrant Ulcers: Factors Prolonging Healing

Acute wounds physiologically heal within 4 weeks; on the other hand, chronic wounds need a longer time to close, with an average healing time of 6–12 months for VLU. Moreover, it has been estimated that recurrence occurs in about 70% of VLU within 5 years of closure [131]. 

Several events occur to delay wound closure, including ulcer characteristics, concomitant diseases, patient characteristics, diagnostic delays and inaccuracies, therapeutic interventions, and environmental factors. 

A well-known risk factor for recalcitrant ulcers is advanced patient age. In fact, elderly patients generally have reduced mobility as well as lesser compliance for compression bandages and garment treatments compared to younger patients. The VLU microenvironment is also characterized by a compromised cellular and biochemical machinery, where senescent fibroblasts fail to respond to proliferative stimuli [132,133,134]. 

Preexisting or underlying venous diseases, including all anatomic levels of venous system disease or deep venous thrombosis, are among the major risk factors for delayed healing.

Patients with higher body mass index (BMI > 25 kg/m^2^) and nutritional deficiencies also have a poor healing prognosis [17].

Larger wound area and longer duration have been reported as clinical signs of poor healing, while data on ulcer location and shape showed contradictory results [135]. Conflicting results regarding also the volume of exudate, the type and amount of wound infection, and the presence of previous ulceration as potential risk factors for prolonged healing may be important factors in delayed VLU healing [135].

Additionally, a history of venous ligation or vein stripping, a history of hip or knee replacement surgery, ankle brachial pressure index < 0.8, and the presence of fibrin covering greater than 50% of the wound area have been associated with prolonged healing [136]. 

Among the environmental conditions predisposing for delayed wound healing, it has been reported that colder temperature was associated with increased risk of ulcer development [137]. 

The diagnostic delays and a misdiagnosed VLU will result in extended time for healing due to delays in proper diagnosis and treatment. This could be further exacerbated if an improper treatment is initiated for the misdiagnosis.

In this respect, additional factors could be examined to improve the diagnostic process, such as biochemical and molecular parameters which affect VLU progression from development to closure or chronicity. 

Biochemical and biomolecular markers of wound healing could be assessed both through wound tissue biopsies (e.g., wnt signaling pathway, β-catenin, c-myc, growth factors, proteases, and miRNA [56,138]) and through soluble biomarkers circulating in the blood or released within the wound fluid (e.g., MMPs, cytokines, growth factors, levels of albumin, and total protein, etc.) [20,75,102,127,139,140,141], which generally represent crucial regulators of tissue remodeling.

In a recent retrospective cohort study involving 65 patients that underwent several treatments for 1 year, it has been demonstrated that, besides the known risk factors (i.e., deep venous disease and post-thrombotic etiology), novel risk factors, such as depression and race (nonwhite), emerged as important factors for VLU development [142]. 

Recently, a tool has been developed to predict the risk of failure to heal VLUs in 24 weeks by taking into account several factors, including patient characteristics (age, history of deep vein thrombosis in the affected leg, calf circumference, compression treatments, and behavioral factors) and ulcer characteristics (duration, area, presence of necrotic tissue, and ulcer area reduction in 2 weeks) [52]. This may prove useful in clinical practice to set treatment goals and patient–provider expectations. 

Similar approaches, taking into consideration the main risk factors for poor VLU healing, can be considered easy-to-use aids to discern patients with a high risk of delayed healing and to assist clinicians during selection of the best therapeutic approach.

### 3.2. Latest Innovations in Surgical Treatment and Drug Therapies 

A number of treatment options have been utilized for patients with VLU in order to promote healing. A key and consistent treatment for VLU is compression therapy that can achieve dynamic pressures of over 60 mm Hg to enable changes in hemodynamics and promote healing [1,143,144]. There are surgical treatments consisting of open surgery involving abolishing venous reflux in the superficial and perforator systems that have importance in healing and preventing VLU [1]. Recently, the EVRA trial involving treatment of the superficial system in patients without any deep venous obstruction were found to have a significant healing benefit in patients undergoing endovenous ablation utilizing a variety of modalities including thermal energy and nonthermal endothelial injury to occlude the axial reflux. It is important to note that the long-term outcomes of the EVRA trial showed reduced rates of recurrence at 3 years. Moreover, this randomized trial demonstrated that, compared to compression alone, compression plus ablation of the superficial reflux decreased time to healing with a mean of 56 days vs. 82 days with only compression [145]. There was a benefit of an ulcer-free interval and healed VLU at 24 weeks, with an 89% probability that early venous intervention is cost-effective over 1 year [146]. However, VLU recurrences are still a major concern, and in this short 24-week period of follow up, between 11 and 16% had a recurrence with no difference in the treatment arm. Perforator surgery has been advocated as a means for expediting VLU healing. A recent Cochrane review evaluating 4 RCTs ( Randomized Controlled Trials) including 332 patients undergoing subfascial endoscopic perforator surgery (SEPS) compared to other treatment modalities (compression and superficial vein surgery) determined that, although SEPS with compression appeared to have benefit at 24 months in VLU healing, the data was low quality with significant risk for bias. Overall, SEPS did not demonstrate a clear benefit in VLU healing due to low and very low quality of evidence, small sample size, and bias [147]. Outflow obstruction of the venous system can lead to post-thrombotic syndrome and VLU. In a large single-center series, 982 obstructive venous outflow lesions were evaluated in 870 patients undergoing iliac venous stents. There were no mortalities, the early thrombotic rate was 1.5%, and the contralateral iliac vein thrombosis was 1%. At 5 years, 62% of patients were pain free, 32% were edema free, and 58% of VLU was healed [148]. Importantly, if the patient with VLU had both outflow obstruction and superficial great saphenous insufficiency, both could be treated in the same setting with excellent results and a VLU healing rate of 64% at 48 months [149]. Noteworthy, endovenous treatments both for the superficial venous system but especially in the deep outflow iliac venous system should be routinely evaluated in patients with VLU and aggressively treated to relieve venous hypertension and to establish outflow patency, with 5-year VLU healing of about 75% [150]. A recent meta-analysis in the treatment of chronic venous outflow obstruction including 12 studies demonstrated a favorable rate of overall VLU healing of 72.1%. The VLU healing was higher for non-thrombotic than thrombotic lesions for the iliofemoral venous system (86.9% vs. 70.3%, respectively, *p* = 0.0022) [151].

Despite compression treatment, and surgical and endovascular venous treatments, the overall healing and recurrence rates for VLU can be as high as 70%. The explanation is multifactorial, including compliance with compression and treatment, procedural failures, mixed VLU disease that encompasses arterial component, incorrect diagnosis of ulcer, and progression of venous disease. However, a key reason is the poorly understood pathophysiology, molecular pathways leading to tissue injury, persistent inflammatory responses and monocyte/lymphocyte-endothelial activation, and oxidative stress. A number of pharmacologic agents including flavonoids, diosmin-based drugs, pentoxifylline, and sulodexide have been tested in RCTs as adjuvant treatments to compression for improving VLU healing. Overall, the data suggest improved healing with vasoactive pharmacologic drugs, but the studies are short, the length of treatment to prevent recurrence is unclear, they are heterogeneous, and they lack patient-reported quality of life outcomes [1]. 

Several biologic products such as bilayer living skin construct (human skin equivalents), fibroblast derivatives, and extracellular matrices and non-biologic products such as poly-N-acetyl glucosamine have been tested in VLU clinical trials and demonstrated benefit [11,12,152,153,154] Although healing is improved with these products with the addition of compression, there are however significant limitations in these trials, and most have short-term follow-up times of less than a year, lack clear evaluation of the venous system to determine if the disease process is primary or secondary venous insufficiency, and do not include important patient-reported outcomes and cost analysis. Recently, there have been development of new products with improved understanding in the pathophysiology of VLU. The placental membranes amnion/chorion allografts have interesting properties for wound healing. These fall under the human cellular repair matrices and are composed of cryopreserved native placenta without an immunogenic trophoblast layer. They have collagen-rich extracellular matrix proteins, providing growth factors, glycosaminoglycans, fibroblasts and epithelial cells, and important mesenchymal stem cells. In vitro placental membranes promote cellular adhesion and migration, cell differentiation, and proangiogenic anti-inflammatory activities and protect growth factor function [155,156]. One randomized controlled trial evaluated dehydrated human placental allograft (dehydrated human amnion chorion membrane, dHACM) in patients with VLU [157]. In a multicenter randomized controlled trial, 84 VLU patients enrolled and were distributed as the study group consisting of 53 patients assigned to placental allograft with multi-layer compression and a control group with 31 patients with multi-layer compression only. The primary outcome measure was 40% wound closure at 4 weeks. The dHACM (one or two applications) group at 4 weeks had a greater percentage of VLU achieving 40% closure (62% vs. 32%, *p =* 0.005) [158]. Complete healing of VLU and the ulcer free interval, which is an important parameter to the patient and which the study did not evaluate, would be important to know. A more recent randomized trial comparing dHACM with compression and a control group with compression determined that, at 16 weeks, the placental derivative group had significantly improved rates of healing (71% versus 44% at 16 weeks, *p* = 0.0065) and a decreased healing time [159]. Further clinical trials are needed to assess ulcer recurrences, cost effectiveness and analysis, and patient-reported outcomes. Another interesting area of study and potential for VLU healing is the targeting of connexins. Connexin 43 gap junction proteins regulate small molecule signaling to and between cells. They have been associated with regulation of inflammatory cytokine release, mediators of fibrosis pathways, and control growth factor response at the cellular level. Importantly, connexin 43 is abnormally upregulated at wound edges of chronic non-healing wounds and VLU [160,161]. The ACT1 peptide is a competitive inhibitor of connexin 43 (a peptide mimetic of the connexin 43 carboxyl terminus), and the application of ACT1 accelerates wound healing in animal models. An interesting phase 2 study included 92 patients with VLU randomized to either the group with ACT1 topical and debridement with four-layer compression bandages or to the group with debridement and four-layer compression bandages. The primary endpoint to this study was mean percent area reduction at 12 weeks, and the secondary endpoint was 100% closure at 12 weeks. The follow-up was up to 6 months. Both the primary and secondary endpoints were in favor of the VLU treated with ACT1 (79% vs. 36%, *p =* 0.02; 57% vs. 28%, *p =* 0.01; respectively). The VLU recurrence rates at 6 months were equal for each group (11%) [158].

Peroxynitrite (ONOO), a product of nitric oxide and superoxide, is a potent oxidizing and nitrating agent that causes significant and irreversible damage to tissues and cellular components including mitochondria, DNA, lipid peroxidation, posttranslational modifications of many proteins, protein oxidizer and nitration, and enzyme inactivation. ONOO decreases the function of superoxide dismutase (SOD); increases reactive oxygen species (ROS) production, prostacyclin synthase for PGI2 production, glucocorticoid receptor, and response to glucocorticoids; increases COX-2; and activates MMPs. Recently, a very elegant study assessed the presence of ONOO in VLU. Nitrotyrosine is a byproduct indicative of peroxynitrite activity, and poly(ADP-ribose) is the product of the DNA damage sensor enzyme PARP-1. In a study of VLU biopsies compared to normal tissue, the authors found elevated nitrotyrosine and PAR, indicating peroxynitrite oxidation and DNA damage/repair, respectively [104]. These findings confirm that peroxynitrite is present in VLU and likely a significant contributor to pathology of the inflammatory state, and further work in targeting production or activity of ONOO may have significant implications in healing VLU. Innate immunity involving polymorphonuclear cells, macrophages, natural killer T cells, complement system, and lactoferrin are important measures to mitigate infection and to promote wound healing. An important set of molecules are the function of TAM (Tyro, Axl, and MerTK), which is a family of receptor tyrosine kinases and their ligands Gas6 and Protein S (ProS). This group of proteins has innate immune regulation function, is central in the intrinsic inhibition of inflammation to pathogens, and is important in phagocytosis and apoptosis [96]. In a study evaluating the gene expression of patients with VLU (n = 67) vs. controls (n = 42), the blood polymorphonuclear cells were assayed for TAM and their ligands. The TAM and ligands were increased significantly over the control, but importantly, when comparing VLU responders that were healing with VLU non-responders, the responders had significantly elevated TAM Axl elevation while non-responders had significantly elevated Gas ligand [96]. These finding not only are important in defining the role of innate immunity in VLU but has markers for healing progression and targets for potential therapy. 

The toll-like receptor family is important in innate immunity and pattern recognition. These receptors are expressed on the cell surface of innate immune cells and non-immune cells of the dermis and epidermis. They recognize discrete pathogen molecular patterns as well as endogenous damage-associated molecular patterns released after tissue and cellular damage. Importantly, toll-like receptors trigger proinflammatory responses and cytokine release, which are important mechanisms in VLU pathology. In a study assessing VLU wound fluid, toll-like receptors were assessed in healing and non-healing VLU. Both toll-like receptors 2 and 4 were significantly elevated in non-healing VLU and decreased in healing VLU. In addition, the antibacterial peptide lipocalin-2 was elevated in non-healing VLU because of the increased inflammation [91]. The possibility for future research in the areas of innate immunity, modulating pathways, and targeting certain receptors and ligands has significant promise in novel treatment and pharmacology. 

A new and exciting area of research is metabolomics. Metabolic phenotyping has been employed to explore mechanistic pathways involved in venous disease. Metabolomics evaluates both aqueous and non-aqueous metabolites utilizing nuclear magnetic resonance spectroscopy (NMR) and mass spectrometry (MS). A recent systematic review regarding the study of metabolites in VLU determined that upregulated metabolites in wound fluid and ulcer biopsies including lactate, branch chained amino acids, lysine, 3-hydroxybutyrate, and glutamate were identified and have importance in cell energy, amino acid and protein biosynthesis, and cellular functions. These data provide important clues to the disease pathophysiology within VLU, and further research on mechanisms and targeted therapy hold significant promise [14]. 

Sulodexide is a glycosaminoglycan with the composition of 80% heparan sulfate (also known as fast-moving heparin) and 20% dermatan sulfate. The heparan sulfate congener is predominantly composed of glucuronic acid linked to glucosamine, while the dermatan sulfate consists of iduronic acid and galactosamine. Sulodexide has important biologic effects with antithrombotic, anti-inflammatory, and endothelial protective properties [106,162,163]. Although sulodexide was demonstrated nearly two decades ago to be effective in increasing VLU healing [164,165,166], its novel molecular mechanisms and the pleiotropic effects are just recently understood. It is important to assess the effects of the drug in patients with VLU and healed ulcer disease, which provides insight into the mechanisms and targets of sulodexide. In a study evaluating the anti-inflammatory effects of sulodexide in healed VLU patients who were treated for 8 weeks (2 × 500 LSU/day, oral), blood samples before and at completion of the study were drawn to assess for IL-6 and MMP-9. At 8 weeks of treatment with sulodexide, there was a significant decrease in both inflammatory molecules. In addition, evaluation of endothelial cells treated with sulodexide-treated serum from patients significantly decreased IL-6 and intracellular free radicals. Taken together, these data demonstrate that sulodexide results in a reduction in intravascular inflammation and is endothelial-protective [167]. In another interesting study evaluating the serum from CVD-healed ulcer patients before and after treatment with sulodexide, the effect of inflammation and oxidative stress in HUVEC cells was evaluated. The key findings of this study were that sulodexide reduces inflammatory mediators in CVD serum (IL-6, MCP-1, and ICAM-1), reduces oxidative stress, suppresses the effect of IL-1, and reduces population doubling time and hypertrophy, indicating decreased aging and senescence [168]. Further identifying the multiple effects of sulodexide on cellular function, an elegant study assessed sulodexide’s protective action on cell stress and autophagy (a complex process involving lysosomal catabolic actions by which cells degrade or recycle their contents of unnecessary or dysfunctional cellular components to maintain cellular homeostasis, to adapt to stress, and to respond to disease (a protective mechanism). HUVEC cells were stressed metabolically (methylglyoxal) and non-metabolically (ionizing radiation) with and without the treatment of sulodexide. The important and novel findings were the effects of sulodexide mitigating apoptosis by inhibiting intrinsic and extrinsic caspase pathways and increased cell viability, by reducing ROS, by reducing the synthesis and release of inflammatory cytokines (TNF-α, IL1, IL6, and IL8), by promoting cell autophagy in maintaining cellular function, and by reducing DNA damage [169]. The implications are that sulodexide prevents endothelial dysfunction and injury that may have significant implications in CVD, DVT, and PTS. Importantly, these scientific discoveries allow for further research in the pathophysiology of VLU and the possibility for synergistic effects with sulodexide in treating and healing VLU. In another area of recent investigation, sulodexide physiologic properties and effects on MMPs was tested in a murine vein stretch model. Sulodexide caused venous contraction and restored venous contraction in a stretch model compared to an untreated stretched vein while inhibiting MMP-2 and MMP-9 expression and activity, thereby enhancing venous contraction. Sulodexide’s effect on venous contraction was partly due to an increase in the sensitivity of the α-adrenergic receptor, but likely, sulodexide also enhanced downstream mechanisms (Ca^2+^ sensitivity and mobilization, PKC, MAPK, and Rho-kinase). These novel mechanism in SDX vein contraction may be important not only in venous leg ulcers but also in treating CVD in general and improving venous function [170,171].

Several growth factors have been applied to VLU given the importance of many growth factors in the biology of wound healing. These growth factors include platelet-derived growth factor, transforming growth factor, epidermal growth factor, keratinocyte growth factor, and fibroblast growth factor. A recent meta-analysis evaluating 10 studies consisting of 472 in the intervention group (growth factors) and 330 as control determined that there was moderate bias in the study design and that, although tendencies toward healing VLU were present, none reached statistical significance [172]. Further studies with sound methodology, frequency, and duration of growth factor application and proper control are required before strong recommendations can be made. 

Silver dressings have been applied to treat VLU. Silver ions have antibacterial and anti-inflammatory properties, have nonspecific MMP inhibition, and drive senescent cells toward apoptosis. A recent meta-analysis of 8 studies found that, overall, there was a benefit toward VLU healing rate; however, there were no clear benefits in complete VLU healing, and long-term follow-up and comparisons to other wound dressing was lacking, and strong recommendations for general use were not advisable until further research is generated [173]. Another systematic review evaluating the effect of MMP inhibition determined that, from 16 studies utilizing collagen-based and lipido-colloid nano oligosaccharide factor dressings, there was a clear benefit in a variety of ulcers including diabetic, venous, and mixed origin. The major outcomes focused on wound closure, wound size reduction, healing time, and healing rate [174]. Further studies are required to determine which MMPs to target, to focus on VLU etiology and RCT trials, and to determine ulcer free intervals and cost analysis as well as specific biomarkers (e.g., reduction in gelatinase activity in the wound bed) that demonstrate positive progress. It is important to note that topical agents applied to VLU are abundant. However, the data is usually from small studies, with methodology flaws and bias. Many of the studies are of moderate to low certainty and quality of evidence, and further well-defined RCTs with clearly defined inclusion criteria and endpoints that evaluate ulcer free interval, patient-reported outcomes, and cost effectiveness are needed before best medical practices can be offered to patients with VLU and can have the best healing potential and value for the patient [175] (The different treatment strategies and evidence for success in VLU healing have been summarized in Figure 3). 

Bone Marrow derived cell (BMDC) therapy has become an area of intense research in regenerative medicine therapy. A recent pilot study evaluating the feasibility and safety of utilizing BMDC in VLU was conducted. The study included four patients with 6 VLUs. The bone marrow was harvested, processed, and injected in the periulcer bed. At 12-month follow-up, the wound area had decreased and pain had improved [176]. Another pilot trial evaluating progenitor cells obtained from adipose tissue was conducted in 8 treated and 8 control VLU patients. The patients treated with their own progenitor cells had significantly decreased time for healing compared to the control (17.5 ± 7.0 weeks vs. 24.5 ± 4.9; *p* < 0.036) and decreased pain but no difference in rate of healing at 6 months. There were no adverse events [177]. Although promising given the potential nature of BMDC and progenitor cells, future studies require larger RCTs, proper outcome measures, patient-reported outcomes, cost effectiveness, and substantial follow-up to assess for recurrences and any significant adverse events. 

### 3.3. Approaches to Prevent Ulcer Occurrence and Recurrence

Healing VLUs is a significant achievement, but finding lasting treatments that prevent at-risk patients from ulcer formation and recurrences is an important area of study. The Edinburgh Vein Study is a population-based study that randomly selected individuals from 18 to 64 years of age and followed them longitudinally. In assessing the progression of CVI (defined as the presence of skin changes) from their cohort of subjects, the authors found some very interesting epidemiologic characteristics. Of the original 1566 subjects screened, 880 had follow-up examinations and 334 had CVI or varicose veins at baseline. This latter group composed the study sample for evaluating progression. The mean duration of follow-up was 13.4 ± 0.4 years, and progression was determined in 57.8%, resulting in an annual progression rate of 4.3%. In 270 (80.8%) subjects with only varicose veins at baseline, 32% developed CVI with skin changes that significantly increase the risk of VLU. Individuals with combined varicose veins and CVI at baseline were at high risk for progression (98.2%). The annual rate of CVI was 2.6%, and having baseline CVI was associated with development of VLU. Significant risk factors for progression of CVD was varicose veins at baseline and family history of DVT. Obesity was not found to be an independent risk factor for progression of CVI, although this may be a factor and may be implicated since obesity is associated with varicose veins. However, when evaluating subjects older than 55, obesity was a significant factor for the development of CVI. Superficial reflux in the venous system, especially in the small saphenous vein, is an independent risk for CVI progression, especially when combined with deep venous reflux [178]. An important question to ask would be if early intervention by treatment of the venous system and/or compression can prevent the progression of CVD and VLU formation. In addition, utilization of ultrasound to evaluate venous reflux in both the superficial and deep venous systems could be an important tool for prevention and prognosis, instituting aggressive treatment with compression, venous surgical correction of reflux, and the use of pharmacologic venous drugs (e.g., diosmin, pentoxifylline, and sulodexide). Preventing ulcer formation and susceptibility is of paramount importance. Identifying individuals at risk for progression of CVI and VLU by history of varicose veins or DVT along with a complete investigation of the venous system with ultrasound and identifying reflux that is correctable may prevent the formation of de novo VLU [179]. Genetic predisposition may also be a significant factor for disease progression and VLU development. Currently, it is unclear if genetic alterations in a number of identified genes are causal or associations. Certainly, this area of research is interesting and requires further study in larger population-based studies and determination that early intervention in identified and affected individuals with specific venous gene polymorphic variants would benefit from interventions to correct venous hemodynamics and prevent VLU formation [180,181,182,183,184]. Important principles to prevent and reduce recurrent VLU are following adopted principles of timely referral to a vascular specialist, evaluation and ultrasonography, treating infection and debridement, appropriate compression, and appropriate interventions when indicated [1]. It is important to note that, in patients with healed and active VLUs, 50% stenosis of iliac venous outflow obstruction can be present in up to 37% of individuals and 80% stenosis of iliac venous outflow obstruction is present in 23%. Significant risk factors for venous outflow obstruction include previous DVT, deep venous insufficiency, and female gender [185]. It is important that properly selected individuals with healed or active VLU are assessed for venous outflow obstruction and that, if obstruction is observed that proper treatment with venography, intravascular ultrasound, angioplasty, and stents are offered [186]. Adequate compression (multi-layer bandages, short-stretch bandages, and inelastic and elastic compression) is of primary importance in healing VLU and has ample high levels of evidence [1,187]. What is less certain is the effect of compression in preventing VLU recurrence. The strength of the available data is less clear and inconsistent with unequivocal benefit due to study design, strength and type of compression used, and heterogeneity. However, overall, the data appears favorable, and in clinical practice, clinicians advise and prescribe compression therapy for patients with healed VLU to maintain hemodynamics and to prevent recurrence, which has also been demonstrated to be cost-effective [1,188,189,190,191]. Currently there is insufficient data to determine if compression therapy prevents occurrence and progression of CVD to VLU [191]. Although several pharmacologic vasoactive drugs have been studied and demonstrated to help VLU healing as an adjunct to compression, it is unknown if continued treatment with these agents has an effect on reducing VLU occurrence or recurrence [192]. The EVRA trial demonstrated that surgical intervention on the superficial system decreases the time for VLU healing (median time to healing compression + endovenous ablation 56 days vs. compression 82 days). Noteworthy, the EVRA trial demonstrated that early endovenous ablation of superficial venous reflux was highly likely to be cost-effective over a 3-year horizon compared with deferred intervention, suggesting that early intervention accelerated the healing of venous leg ulcers and reduced the overall incidence of ulcer recurrence. Therefore, the long-term outcomes of the Early Venous Reflux Ablation (EVRA) randomized trial showed accelerated venous leg ulcer healing and greater ulcer-free time for participants who are treated with early endovenous ablation of lower extremity superficial reflux. In addition, the healing rate at 6 months was significantly better for the intervention group at 85.6% vs. 76.3% for compression alone. However, VLU recurrences at 1 year were between 11% and 16%, with no difference in treatment groups, and both groups had improved quality of life after treatment [145,193,194]. An interesting study demonstrated that overall VLU recurrence was 29% at 3 years following venous surgical interventions. However, at 24 months, patients undergoing superficial vein ablation with concomitant phlebectomy (surgical removal of varicose vein tributaries) had a significantly lower recurrence than the group that only had ablation of the superficial venous system (12% vs. 24%, respectively). In addition, having deep venous insufficiency was also a predictor of VLU recurrence [193]. Reduction in VLU recurrence was also demonstrated at four-year follow-up in the ESCHAR trial in patients with VLU undergoing saphenous venous surgery as an adjunct to compression compared to compression alone (31% vs. 56%, respectively, *p* ˂ 0.01) [194]. 

It is important to note that the benefits of intervention and that the majority of VLU are persistent due to a lack of referral to a venous specialist.

Figure 4 provides the treatments and areas of needed study fields that affect VLU occurrence and recurrence. 

## 4. Conclusions and Future Perspectives

Venous leg ulcer (VLU) is a complex lower extremity disorder associated with post-thrombotic syndrome and/or advanced CVD, primary venous insufficiency and varicose veins, and venous hypertension. Compared to other ulcers, VLU is the most common ulcer of the lower extremity. VLU affects a significant portion of the population in the western, eastern, and developing worlds, with high incidence in the United States, the United Kingdom, Europe, Australia, India, and Africa. The increasing cost of medical care for VLU poses a significant financial and socioeconomic burden to the healthcare system. VLU could also have major emotional, psychological, and physical impacts on the affected individual. Several predisposing demographic, genetic, and environmental factors could lead to CVD with extensive venous dilation, incompetent valves, venous reflux, and venous hypertension. Endothelial cell injury, damage to the endothelial glycocalyx, increased adhesion molecules, and changes in venous shear-stress could also be major initiating events in VLU. Increased endothelial cell permeability, leukocyte infiltration, and inflammation; increases in inflammatory cytokines, MMPs, and reactive oxygen and nitrogen species; and accumulation of iron deposits and tissue metabolites could also contribute to the pathogenesis of VLU. VLU usually heals with good wound care and compression therapy within 6 months. The VLU healing process involves multiple steps including hemostasis, inflammation, cell proliferation, and tissue remodeling. VLU healing also involves the contribution of multiple cell types including leukocytes, platelets, endothelial cells, vascular smooth muscle cells, fibroblasts, and keratinocytes. Additionally, VLU healing involves the release of a wide variety of localized and circulating biomolecules including TGF-β, VEGF, PDGF, TNF-α, interleukins, chemokines, MMPs, TIMPs, elastase, urokinase plasminogen activator, fibrin, collagen, albumin, and other proteins. Alteration in any of these cellular and biochemical components that drive the physiological wound healing process could delay VLU healing. Importantly, while histological studies are highly informative, specimens are obtained through invasive procedures that reflect a single time point during VLU healing, making it difficult for repeated sampling from the same site. On the other hand, soluble biomarkers in the blood and wound exudate can be measured with less invasive techniques and at different stages of VLU progression. Also, instead of relying solely on measurement of the VLU area to evaluate the wound healing status, these VLU healing factors could be used as biomarkers for ulcer diagnosis and for the evaluation of its progression, responsiveness to healing, and prognosis.

Inadequate treatment of VLU could lead to progression to non-healed or granulating VLU with major complications including physical immobility, reduced quality of life, cellulitis, severe infections, osteomyelitis, and neoplastic transformation. Recalcitrant ulcers also show prolonged healing time and often occur in individuals with advanced age, higher body mass index, and nutritional deficiencies and in association with colder temperature, preexisting or underlying venous disease, deep venous thrombosis, venous outflow obstruction, and larger wound area. 

Treatment of VLU includes compression therapy and endovenous ablation utilizing a variety of modalities including thermal energy and non-thermal endothelial injury to occlude the axial reflux. Other interventional approaches include endovenous treatments for both the superficial venous system using subfascial endoscopic perforator surgery (SEPS) and the deep outflow iliac venous system using iliac venous stents, but the beneficial effects of these approaches need to be further evaluated. Importantly, VLU has a high 50–70% recurrence rate possibly due to patient noncompliance with compression therapy, surgical procedure failure, mixed VLU disease with arterial component, incorrect diagnosis of ulcer, and progression of venous disease. A key reason for VLU recurrence is also poorly understood pathophysiology and the molecular pathways leading to tissue injury, persistent inflammatory responses and monocyte/lymphocyte-endothelial activation, and oxidative stress (Figure 5).

New lines of therapy for VLU have been tested or are being developed. Several biologics such as bilayer living skin construct, fibroblast derivatives, and extracellular matrices and non-biologic products such as poly-N-acetyl glucosamine have been tested and demonstrated benefits in VLU clinical trials. Human placental membrane amnion/chorion allografts have interesting properties for wound healing as they have collagen-rich ECM proteins and could provide growth factors, glycosaminoglycans, fibroblasts, epithelial cells, and mesenchymal stem cells necessary for healing. 

Connexin 43 is abnormally upregulated at the wound edges of chronic non-healing VLU, and the ACT1 peptide is a competitive inhibitor of connexin 43 that accelerates wound healing in animal models. Also, sulodexide has antithrombotic, anti-inflammatory, and endothelial protective properties that could support VLU healing. Other pharmacologic drugs may also have benefits in VLU healing. Growth factors; bone marrow and adipose tissue-derived cell therapy; silver dressings; MMP inhibitors; and modulators of reactive oxygen and nitrogen species, the immune response, and tissue metabolites could also have benefits in VLU.

Lastly, preventing VLU formation and susceptibility is very important. Identifying individuals at risk of progression to CVI and VLU and treatment of these individuals promptly with compression therapy and venotonics could reduce the incidence and recurrence of VLU and prevent the progression to non-healed and recalcitrant VLU.

## Figures and Tables

**Figure 1 jcm-10-00029-f001:**
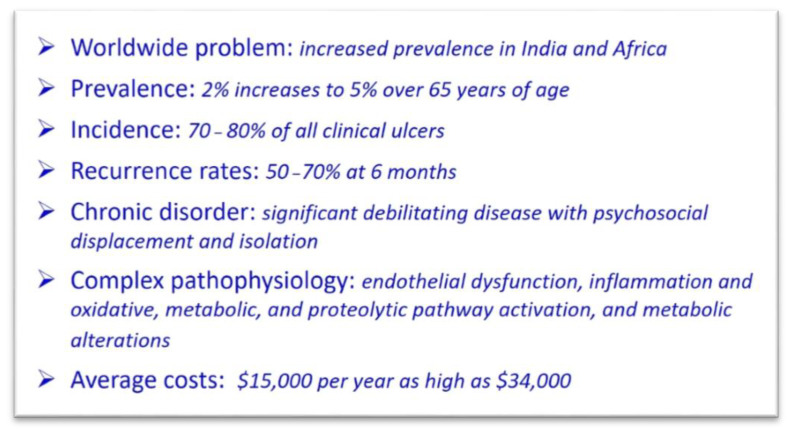
Facts about venous leg ulcer.

**Figure 2 jcm-10-00029-f002:**
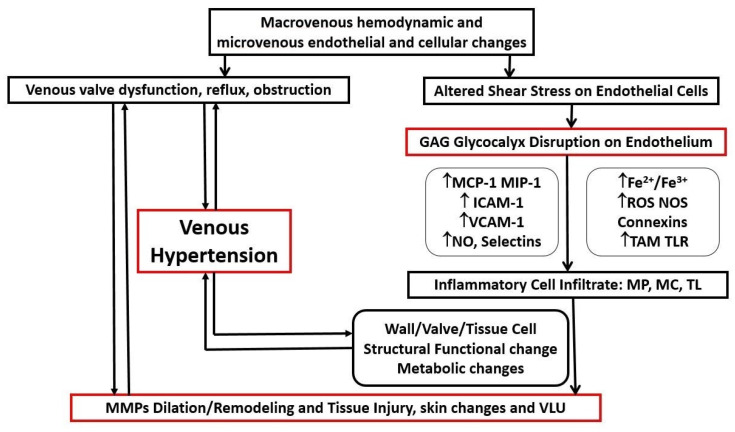
Diagram representation of chronic venous disorder pathophysiology. GAG: glycosaminoglycans, MCP-1: monocyte chemoattractant protein, MIP-1: macrophage inflammatory protein, ICAM-1: intercellular adhesion molecule, VCAM-1: vascular cell adhesion molecule, NO: nitric oxide, Fe^2+^/Fe^3+^: ferrous/ferric ions, ROS: reactive oxygen species, NOS: nitrogen oxidative species, TAM: Tyro Axl MerTK receptor family tyrosine kinase, TLR: toll like receptors (in particular: ↑ increased concentration ↓ decreased concentration).

**Figure 3 jcm-10-00029-f003:**
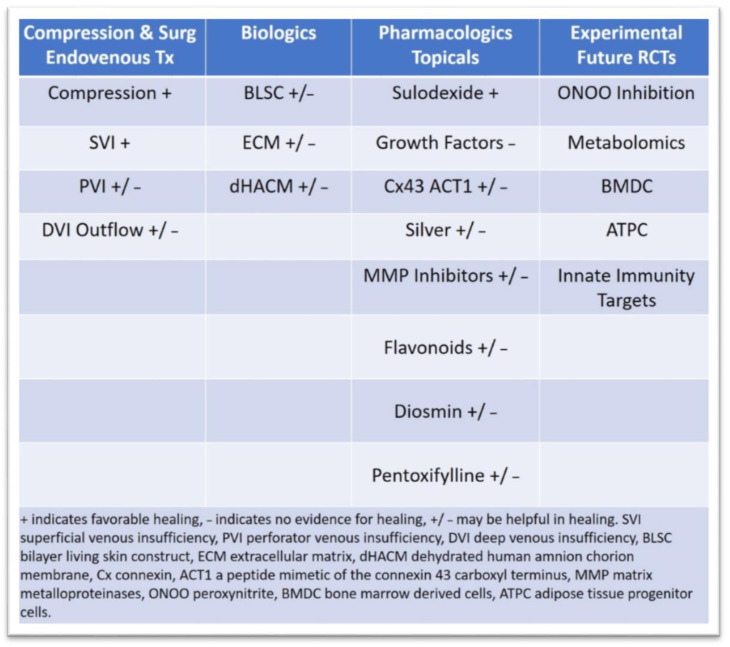
Venous leg ulcer treatments and healing potential.

**Figure 4 jcm-10-00029-f004:**
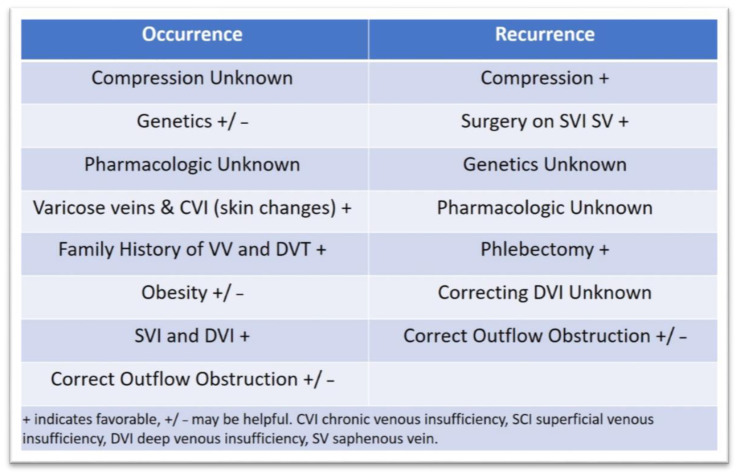
Treatments and study fields that affect venous leg ulcer occurrence and recurrence.

**Figure 5 jcm-10-00029-f005:**
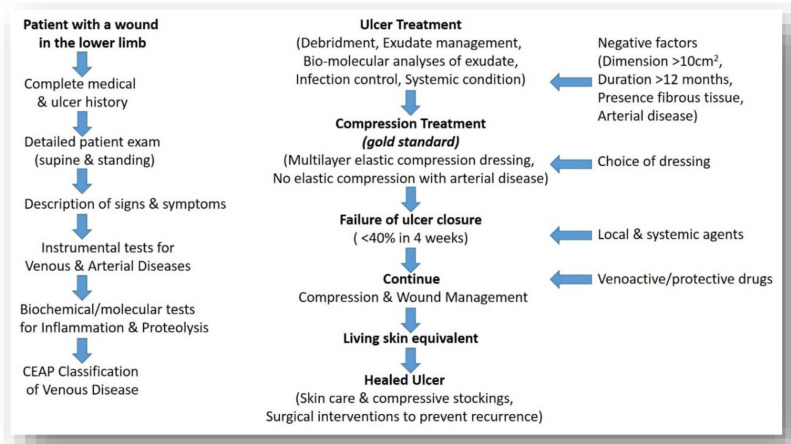
Schematic representation of a possible algorithm for treating venous leg ulcer.

**Table 1 jcm-10-00029-t001:** Comparison among the most diffused leg ulcers of vascular origin.

**Ulcer Type**	**Location**	**Clinical** **Presentation**
Venous ulcer	Gaiter region of the lower leg (anterior to medial malleolus, pretibial lower third of leg, occasionally lateral malleolus)	Single or multiple lesions; shallow depth; irregular shaped edges with well-defined margins; exudates yellow-white in color; commonly with granulation and fibrinous tissue and rarely with necrotic tissue; associated pain may be absent, mild, or extreme; lower extremity edema; eczema and pruritus; hemosiderin deposition or lipodermatosclerosis; dilated and tortuous superficial veins; inverted champagne bottle appearance of the lower leg
Arterial ulcer	Distal extremities and sites of trauma (e.g., over the toes, heels, and bony prominences)	Sharply demarcated borders; base yellow, brown, grey, or black in color and usually does not bleed; pale, dry, non-granulating and often necrotic wound bed; the surrounding skin may exhibit erythema, may be cool to touch, and may be hairless and thin; substantial pain, often severe, worsens in decubitus position or when walking; intermittent claudication (leg pain with exercise or at rest); toe nails become opaque and may be lost or hypertrophic; gangrene of the extremities may occur; reduction of capillary refill time; low exudate unless ulcers are infected
Lymphatic ulcer	Frequently in the ankle area but may develop in the trauma sites	Shallow depth; regular shaped; flat edge; rosy base; may be oozing, moist, or blistered; lymphorrhea; edema with buffalo hump on the dorsum of the foot and a positive Stemmer’s sign; the skin is translucent, cold, pale, unpigmented, and rarely fibrosclerotic.
Vasculitic ulcer	Multifocal or atypical areas.	Sharply marginated; ulcers can be single or multiple with necrosis and fibrin congestion; morphology depends on the size of the vessels and extent of the vascular bed affected; usually fever, weight loss, fatigue joint pain, and rash; reticulated erythema; widespread purpura; the skin surrounding ulcer is normal both before and after ulcer development; painful ulcer
Atypical ulcers	Cutaneous and characterized by an atypical wound bed, edges, and perilesional skin: the clinical aspects are correlated with different etiologies.	The wound bed is often exuberant or vegetative, with hyper-granulation tissue or necrotic tissue. Wound edges are undermined or exuberant. Perilesional skin may present with inflammation or satellite lesions. They are caused by inflammatory, neoplastic, vasculopathic, hematological, infectious, and drug-induced etiologies. Approximately 20% of these ulcers are caused by rare etiologies.
**Ulcer Type**	**Location**	**Clinical Presentation**
Venous ulcer	Gaiter region of the lower leg (anterior to medial malleolus, pretibial lower third of leg, occasionally lateral malleolus)	Single or multiple lesions; shallow depth; irregular shaped edges with well-defined margins; exudates yellow-white in color; commonly with granulation and fibrinous tissue and rarely with necrotic tissue; associated pain may be absent, mild, or extreme; lower extremity edema; eczema and pruritus; hemosiderin deposition or lipodermatosclerosis; dilated and tortuous superficial veins; inverted champagne bottle appearance of the lower leg
Arterial ulcer	Distal extremities and sites of trauma (e.g., over the toes, heels, and bony prominences)	Sharply demarcated borders; base yellow, brown, grey, or black in color and usually does not bleed; pale, dry, non-granulating and often necrotic wound bed; the surrounding skin may exhibit erythema, may be cool to touch, and may be hairless and thin; substantial pain, often severe, worsens in decubitus position or when walking; intermittent claudication (leg pain with exercise or at rest); toe nails become opaque and may be lost or hypertrophic; gangrene of the extremities may occur; reduction of capillary refill time; low exudate unless ulcers are infected
Lymphatic ulcer	Frequently in the ankle area but may develop in the trauma sites	Shallow depth; regular shaped; flat edge; rosy base; may be oozing, moist, or blistered; lymphorrhea; edema with buffalo hump on the dorsum of the foot and a positive Stemmer’s sign; the skin is translucent, cold, pale, unpigmented, and rarely fibrosclerotic.
Vasculitic ulcer	Multifocal or atypical areas.	Sharply marginated; ulcers can be single or multiple with necrosis and fibrin congestion; morphology depends on the size of the vessels and extent of the vascular bed affected; usually fever, weight loss, fatigue joint pain, and rash; reticulated erythema; widespread purpura; the skin surrounding ulcer is normal both before and after ulcer development; painful ulcer
Atypical ulcers	Cutaneous and characterized by an atypical wound bed, edges, and perilesional skin: the clinical aspects are correlated with different etiologies.	The wound bed is often exuberant or vegetative, with hyper-granulation tissue or necrotic tissue. Wound edges are undermined or exuberant. Perilesional skin may present with inflammation or satellite lesions. They are caused by inflammatory, neoplastic, vasculopathic, hematological, infectious, and drug-induced etiologies. Approximately 20% of these ulcers are caused by rare etiologies.

**Table 2 jcm-10-00029-t002:** Differential diagnosis of leg ulcers.

Ulcer Etiology	Ulcer Type
Vascular	Venous, arterial, lymphatic, vasculitis
Metabolic	Diabetes mellitus, gout, necrobiosis lipoidica, porphiria cutanea tarda, homocysteinuria, prolidase deficiency, hyperoxaluria, ulcerative colitis, avitaminosis, cutaneous calcinosis
Connective tissue disease	Inflammatory bowel disease, pyoderma gangrenosum, rheumatoid arthritis, generalized and localized scleroderma, systemic lupus erythematous, bullous pemphigoid, dermatomyositis, Sjogren’ syndrome, polyarteritis nodosa, leukocytoclastic vasculitis
Cutaneous microthrombocitic ulcers	Cryofibrinogenemia, cryoglobulinemia, antiphospholipid syndrome, coagulopathies, calciphylaxis, cholesterol embolization
Hematological disease	Sickle cell disease, leukemia, thrombocytosis, thalassemia, hereditary spherocytosis, glucose-6-phosphate dehydrogenase deficiency, essential thrombocythemia, granulocytopenia, polycythemia, monoclonal and polyclonal, dysproteinemia
Neoplastic disease	Basal cell carcinoma, squamous cell carcinoma, malignant melanoma, primary cutaneous B cell lymphoma, Marjolin’s ulcer, pseudoepitheliomatous hyperplasia, Kaposi’s sarcoma, angiosarcoma, Bowen’s disease, intra-epidermal carcinoma, papillomatosis cutis carcinoid, neoplasms of lymphoproliferative tissue, Hodgkin disease
Panniculitis	Necrobiosis lipoidica, erythema nodosum, erythema induratum
Traumatic	Pressure ulcers, radiation damage, thermal burns, decubitus
Iatrogenic	Drugs.
Atypical	Cutaneous ulcer, caused by inflammatory, neoplastic, vasculopathic, hematological, infectious, and drug-induced etiologies, with approximately 20% of these ulcers caused by rare etiologies
Martorell HYTILU	Hypertensive ischemic leg ulcer, stenotic subcutaneous arteriolosclerosis
Infection	Pyogenic, osteomyelitis, tuberculosis, syphilis, tropical disease, fungal disease, leishmaniasis, histoplasmosis, herpes, lupus vulgaris, amoebiasis, chromoblastomycosis, coccidiomycosis, viral

**Table 3 jcm-10-00029-t003:** Summary of findings from studies investigating tissue inflammatory biomarkers in human venous leg ulcers.

Main Findings	Specimens	Ref
↑ TNF-α in ulcer vs. normal tissue	Ulcer tissue	[61]
↑ TGF-β1 in ulcer fibrin cuff vs. normal tissue	Ulcer tissue	[62]
No changes of PDGF-AB, GM-CSF, IL-1α, IL-1β, IL-6, and bFGF in non-healing vs. healing ulcers	WF	[63]
↑ IL-6 level; no changes IL-1β, IL-2, and TNF-α in ulcer vs. normal serum	Serum	[64]
↓ TGF-β RII	Fibroblasts from venous ulcer	[65]
↑IL-1ra, IL-6, and PAF in resting ulcer effluent vs. systemic blood; no changes in TNF-α and IL-1β	Blood	[66]
↑ IL-1β, IP-10, and PF4; ↓ IL-1β, MIP-1β, and RANTES; and ↑ IL-1ra, IL-10, MCP-1, and MIP-1α in healing ulcers	Ulcer tissue and WF	[67]
↑ TNF-α and p75 receptor in nonhealing vs. healing ulcers	WF	[68]
↑ EGFR, bFGF, and TGF-β3 in ulcers vs. normal tissue	Ulcer tissue	[69]
↑ PDGFR-α and PDGFR-β, VEGF	Ulcer tissue	[70]
↑ IL-10; no change GM-CSF in ulcers vs. normal tissue	Ulcer tissue	[71]
↑ IL-10 in ulcer vs. normal tissue	Ulcer tissue	[72]
↑ TGF-β1 in ulcer vs. normal tissue	Ulcer tissue	[73]
↑ sThy-1 in UWF vs. serum	WF and serum	[74]
↑ IL-1, IL-6, and TNF-α in non-healing vs. healing; no change in PDGF, EGF, bFGF, and TGF-β	WF	[75]
↑ VEGF in ulcer vs. normal tissue	Ulcer tissue	[76]
↑ TGF-β1, -2, and -3; TGF-β RI; and RII in healing vs. non-healing ulcers	Ulcer tissue	[77]
↑ VEGF and TNF in non-healing vs. healing	Serum	[78]
No changes in TGF-β1 in ulcer vs. normal tissue	Ulcer tissue	[79]
↓ TGF-β RII in ulcer fibroblasts vs. normal tissue	Ulcer tissue	[80]
↑ TNF-α, TNF-rI, IL-1α, IL-6, TGF-β1, PDGF-A, EGF, bFGF, and VEGF in fibroblast from ulcer edge vs. control ↑ PDGF-A and VEGF in non-healing vs. healing ulcers	Ulcer tissue	[81]
↓ IL-8 in healing vs. non-healing ulcer	WF	[82]
↑ c-met in ulcer vs. normal skin ↑ HGF in chronic vs. acute UWF	Ulcer tissueWF	[83]
↑ IL-1α, IL-1β, IL-1ra, EGF, and PDGF-A in endothelial cells near vs. distant ulcer; no changes in IL-6, GM-CSF, and TNF-α	Ulcer tissue	[84]
↑ RANTES mRNA ulcer vs. normal	Blood	[85]
↑ TGF-β1 in healing vs. non-healing	WF and blood	[86]
↑ TGF-β1 and IL-1ra, and ↓IFN-γ in healing ↑ IL-1α, IL-1β, IFN-β, IL-12p40, and GM-CSF in non-healing	Ulcer tissue	[87]
↑ TNF-α in ulcer vs. normal tissue	Ulcer tissue	[88]
↑ RANTES mRNA ulcer vs. normal	Blood	[89]
↑ IL-6 and TNF-α in healed ulcer vs. normal tissue	Valve tissue	[90]
↓ level of IL-8 and MIP-1α in non-healing ulcers ↓ level of IL-1α, IL-1β, and MIP-1δ in healing ulcers	WF	[91]
↓ S100A8/A9 in nonhealing vs. healing	WF	[92]
↑ IL-1α, IL-1β, and IL-8 in WF secreted for 24h vs. WF secreted for 1h	WF	[93]
↑ IL-8, GRO-α, MIP-3α, PARC, HGF, IL-6, MIP-1α, MCP-1, bFGF, TGF-β, CTAK, RANTES, SDF-1, IL-10, and TNF-α	WF	[94]
↑ sVEGFR-1 in non-healing venous wound ↓ VEGFR-2	WFTissue and plasma	[95]
↑ mRNA of TAM receptors and their ligands (Gas6 and ProS) in VLU patients vs. control probands ↑ IL-1α and CXCL-8 gene expression in non-responder vs. responder VLU patients	PBMCs from patients with VLU	[96]
↓ IL-6, IL-8, VEGF, and TNF- α in relation to ulcer healing speed	Plasma	[97]
↑ IL-1, IL-6, IL-8, VEGF, and TNF-α in infected ulcers vs. uninfected ulcers	Plasma and WF	[98]
↑ NGF and S100A8/A9 in painful ulcers	WF	[99]
↑ IL-1β, IL-1ra, IL-6, IL-8/CXCL8, IL-10, IL-12, IL-17, bFGF, G-CSF, GM-CSF, INF-γ, MCP-1/CCL2, MIP-1α,/CCL3, MIP-1β/CCL4, TNF-α, and VEGF ↑ Eotaxin/CCL11, IP-10/CXCL10, and RANTES/CCL5	WFPlasma	[20]
↓ S100A8/A9 in VLUs vs. DFUs ↑VEGF in VLUs vs. DFUs	WF	[100]
↑ PDGF-AA, PDGF-AA receptor, PDGF-BB, and PDGF-BB receptor ↑ TGF-β in injured skin vs. healthy skin	Ulcer tissue	[101]
↑TGF-β3 and soluble endoglin	WF	[102]
↑OPN	Ulcer tissue	[103]
↑LDH activity, IL-8, TNF-α, and VEGF in chronic wound vs. acute wound ↑ Nitrotyrosine and Poly(ADP-Ribose)	WFUlcer tissue	[104]
↑GM-CSF, IRF5, TNF-α, IL-1β, and IL-6 in chronic non-healing ulcers	WF	[51]
↑ MCP-1, IL-1β, IL-4, IL-6, IL-8, MIP-1α, FGF-2, and VEGF-A ↓ G-CSF and GM-CSF	Serum	[105]

(in particular: ↑ increased concentration ↓ decreased concentration).

**Table 4 jcm-10-00029-t004:** Summary of findings from studies investigating soluble proteolytic biomarkers in human venous leg ulcers.

Main Findings	Specimens	Refs
↑ MMP-1 in migrating keratinocytes and superficial dermal cells in chronic compared to acute ulcers	Ulcer tissue and in vitro cell culture	[107]
↑TIMP-1 and TIMP3 in proliferating keratinocytes and ↑TIMP-2 in migrating epithelium in acute compared to chronic wounds	Ulcer tissue	[108]
↑ MMP-1 and MMP-8; ↓ TIMP-1 in nonhealing compared to healing ulcers	Ulcer tissue and WF	[109]
↑ MMP-9 in non-healing compared to healing ulcers	WF	[110]
↑MMP-1 mRNA (no changes in protein) in C4 and C6 stages compared to healthy skin; ↑TIMP-1 mRNA (no changes in protein) in C6; ↑active MMP-2 in C4 and C5 stages	Ulcer tissue	[111]
↑MMP-2 and ↑MMP9 in epithelium/edge of acute wounds compared to healthy skin; MMP2 and MMP-9 localized in ulcer bed	Ulcer tissue	[112]
↑EMMPRIN, ↑MMP2, ↑MT-1MMP, and ↑MT2-MMP in ulcer tissue compared to healthy skin	Ulcer tissue	[113]
↑MMP-7, ↑MMP-12 (epithelium), and ↑MMP-13 in malignant ulcers	Ulcer tissue	[114]
↑MMP-2, ↑MMP-9, and angiogenesis induction by wound fluid from chronic compared to acute wounds; ↓ angiogenesis when MMP-2 and MMP-9 were inhibited	WF	[115]
↑ MMP-9 activation in C4–C6 patients compared to healthy subjects	Serum	[116]
No changes in MMP-2, TIMP-2, and EMMPRIN; ↑ PDGF-AA in healing compared to non-healing ulcers	Ulcer tissue and WF	[117]
↑ total MMP in ulcer tissue compared to healthy skin; ↑ collagen turnover; ↑MMP-1 and no changes in total MMPs and MMP-3 in healing compared to resistant ulcers	Ulcer tissue	[118]
↑ MMP-1, ↑ MMP-3, and ↓ TIMP-1 in fibroblast exposed to wound fluid from chronic compared to acute UWF	WF	[119]
No changes in MMP-9 in relation to ulcer healing	WF and venous blood	[86]
↑ MMP1, 2, 3, 8, 9, 12, and 13 in ulcer tissue compared to normal skin; ↓ MMP-1, -2, -8, and -9 in healing ulcers	Ulcer tissue	[120]
↑ MMP-2 and MMP-9 in ulcer compared to normal tissue	Valve tissue	[90]
↑ MMP-2 and MMP-9 in UWF compared to tissue	WF and ulcer tissue	[121]
↑MMP-9 in ulcers compared to healthy subjects	WF, plasma and ulcer tissue	[122]
↑ MMP-1 and MMP-8 in patients with infected compared to uninfected ulcers; ↑ MMP-2 and MMP-9 in uninfected ulcers	WF and plasma	[97]
↓ MMP-2 and MMP-9 in correlation to ulcer healing	WF	[123]
↓ MMP-9 and NGAL in high-healing ulcer vs. low-healing ulcers	WF and plasma	[124]
↑ MMP-1 and MMP-8 in non-healing wound vs. healing wound	WF and ulcer tissue	[125]
↑ MMP-2, MMP-9, TIMP-1, and TIMP-2 venous leg ulcers vs. healthy controls ↓ MMP-9, TIMP-2, and MMP-9/TIMP-1 ratio in healing ulcers ↑ MMP-2, MMP-9, TIMP-1, TIMP-2, and MMP-2/TIMP-2 ratio in healing ulcers vs. healthy controls	Plasma	[126]
↓ MMP-1, MMP-2, MMP-9, NGAL, and MMP-8 in relation to ulcer healing speed	Plasma	[98]
↑ MMP-1 and MMP-8 in infected ulcers vs. uninfected ulcers ↑ MMP-2 and MMP-9 in uninfected ulcers vs. infected ulcers	Plasma	[97]
↑ MMP-2, MMP-9, MMP-12, TIMP-1, and TIMP-2 in VLU during inflammation ↑ MMP-1, MMP-7, MMP-13, and TIMP-4 in VLU during granulating phases	WF	[127]
↑ MMP-2, MMP-8, MMP-9, and HNE in chronic wound vs. healing wound	WF	[128]
↑ MMP-1, MMP-8, ADAM-17, and ADAMTS-4 ↓ ADAMTS-5, TIMP-1, and TIMP-2	Serum	[124]
↑ MMP-9 in wound fluid vs. corresponding tissue	WF	[129]
↓ TIMP-1 in chronic VLU vs. acute VLU	WF	[130]
↑ MMP-13 in chronic non-healing wounds	WF	[51]

(in particular: ↑ increased concentration ↓ decreased concentration).

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
