# Peer review of "Why Venous Leg Ulcers Have Difficulty Healing: Overview on Pathophysiology, Clinical Consequences, and Treatment"

_jcm, 2020, doi:10.3390/jcm10010029_

Round 1

Reviewer 1 Report

The review is very thorough and carefully written. However, some modifications are suggested:

  1. page 6 r.159-160 as a group should be mentioned Atypical wounds that comprise 10 -20 % of all chronic wounds (Reference: EWMA Document. Atypical Wounds. Best Clinical Practices and Challenges). Accordingly, table 1 and table 2 should be modified and Martorell HYTILU ulcers should be mentioned as they are an important differential diagnosis considering venous leg ulcers.
  2. The authors describe very extensively the biomarkers in venous leg ulcers. The reader is left to think are there any good papers about the effect of compression in these biomarkers, as compression therapy is the golden standard in treatment ( for instance Seidler SK et al Wound Repair Regen 2008 :16(5):642-8
  3. in the end of review, a clear algorith/table about the standard treatmets and the novel possibilities would clarify the message.

Author Response

Replies to Reviewer 1

Reviewer1: The review is very thorough and carefully written.

Authors: We would like to thank the Reviewer for the gratifying words.

Reviewer: However, some modifications are suggested:

  1. page 6 r.159-160 as a group should be mentioned Atypical wounds that comprise 10 -20 % of all chronic wounds (Reference: EWMA Document. Atypical Wounds. Best Clinical Practices and Challenges). Accordingly, table 1 and table 2 should be modified and Martorell HYTILU ulcers should be mentioned as they are an important differential diagnosis considering venous leg ulcers.

Authors: We modified the tables according to the insightful suggestions of Reviewer 1

Reviewer:

  1. The authors describe very extensively the biomarkers in venous leg ulcers. The reader is left to think are there any good papers about the effect of compression in these biomarkers, as compression therapy is the golden standard in treatment ( for instance Seidler SK et al Wound Repair Regen 2008 :16(5):642-8

Authors:  According to the suggestions of the Reviewers, we clarify the crucial role of compression in healing process through the modification of both inflammatory and proteolytic biomarkers, citing in particular some works by Biedler et al.

Reviewer:

  1. in the end of review, a clear algorith/table about the standard treatmets and the novel possibilities would clarify the message.

Authors: In agreement with the Reviewer, we added a new figure depicting algorithm of treatments for VLU

Reviewer 2 Report

Excellent and comprehensive review.

Only a few general comments.

Firstly I think this paper would benefit greatly from being separated into two sections - pathophysiology and then clinical aspects.  At present the paper is too long - 34 pages with 14 pages of references!  The average article appears to be about 11 pages including references.  At present due to the amount of information it is very dense and difficult to assimilate.

Secondly I think it is important to highlight the benefits of intervention and that the majority of VLU are persistent due to lack of referral to a venous specialist.

It is important to include the long term outcomes of EVRA which showed reduced rates of recurrence at 3 years.

However, other than this, this is an excellent article.

Author Response

Replies to Reviewer 2

Reviewer 2: Excellent and comprehensive review.

Authors: We would like to thank the Reviewer for the gratifying words.

Reviewer 2: Only a few general comments.

Firstly I think this paper would benefit greatly from being separated into two sections - pathophysiology and then clinical aspects.  At present the paper is too long - 34 pages with 14 pages of references!  The average article appears to be about 11 pages including references.  At present due to the amount of information it is very dense and difficult to assimilate.

Authors: According to the insightful comment, we divided the ms into two main sections as suggested. For what concerns the lenght of this ms, I agree with the Reviewer that can appear too long but, due to is omnicomprehensive characteristic detailed and programmed during the invitation procedure by the Journal (as Editor of Vascular Medicine section), I believe that this review may represent a milestone and a good landmark for our Readers. In this case, the review will be read and cited as the first omnicomprehensive review in this field. Hope you understand and agree with the original plan of this paper.

Reviewer 2: Secondly I think it is important to highlight the benefits of intervention and that the majority of VLU are persistent due to lack of referral to a venous specialist.

Authors: We agree with the Reviewer and have added a sentence as suggested.

Reviewer 2: It is important to include the long term outcomes of EVRA which showed reduced rates of recurrence at 3 years.

Authors: In full agreement with the Reviewer, we added sentences about the long term outcomes of EVRA.

Reviewer 2: However, other than this, this is an excellent article.

Authors: On behalf of all co-authors, I would like to sincerely and deeply  thank for your gratifying words and appreciations; your comments make all of us very proud and happy.